# Classification Accuracy Improvement for Small-Size Citrus Pests and Diseases Using Bridge Connections in Deep Neural Networks

**DOI:** 10.3390/s20174992

**Published:** 2020-09-03

**Authors:** Shuli Xing, Malrey Lee

**Affiliations:** Center for Advanced Image and Information Technology, School of Electronics & Information Engineering, Chon Buk National University, Jeonju, Chon Buk 54896, Korea; xingshuli600@gmail.com

**Keywords:** citrus pests and diseases, convolutional neural network, bridge connection, additional computation cost

## Abstract

Due to the rich vitamin content in citrus fruit, citrus is an important crop around the world. However, the yield of these citrus crops is often reduced due to the damage of various pests and diseases. In order to mitigate these problems, several convolutional neural networks were applied to detect them. It is of note that the performance of these selected models degraded as the size of the target object in the image decreased. To adapt to scale changes, a new feature reuse method named bridge connection was developed. With the help of bridge connections, the accuracy of baseline networks was improved at little additional computation cost. The proposed BridgeNet-19 achieved the highest classification accuracy (95.47%), followed by the pre-trained VGG-19 (95.01%) and VGG-19 with bridge connections (94.73%). The use of bridge connections also strengthens the flexibility of sensors for image acquisition. It is unnecessary to pay more attention to adjusting the distance between a camera and pests and diseases.

## 1. Introduction

Pests and diseases are major causes of huge economic loss in agricultural production. Timely detection and identification of these pests and diseases is essential to control their impact. Farmers used to rely on experienced experts to perform these tasks; unfortunately, this is a time-consuming, error-ridden, and costly process. To improve recognition efficiency, image processing and computer vision techniques are widely implemented. Bashish et al. [1] proposed a solution for automatic detection and classification of plant leaf diseases. Their method was based on image processing, which used a K-means clustering technique to segment RGB images and then used a single-layered artificial neural network model for classification. Ali et al. [2] presented a citrus disease recognition system in which a ∆E color difference algorithm was adopted to separate the disease affected areas, and a color histogram and texture features were extracted for classification. The features needed to recognize agricultural pests are more complex than those needed for diseases. They are sensitive to affine transformation, illumination, and viewpoint change. Wen et al. [3] introduced a local feature-based identification method to account for variations in insect appearance. Xie et al. [4] fused multiple features of insect species together in order to enhance recognition performance. Compared with other feature-combination methods, their approach produced higher accuracy, but needed more time to train.

The above research all required pretreatment to select features for classifiers. In contrast, deep convolutional neural networks (CNN) are representation-learning models [5]. They can receive raw image data as input and automatically discover the useful features for classification and detection. Various pieces of research were conducted with them, and they achieved satisfactory results. Mohanty et al. [6] employed two pre-trained CNN architectures to classify plant disease images in a public dataset. The deeper GoogleNet performed better than AlexNet in their experiment. The same image dataset was used by Too et al. [7], as they extended the types of CNN model for comparison. Their experimental results showed that DenseNet with more layers got the highest test accuracy. The pre-trained models also play an important role in pest recognition tasks. Cheng et al. [8] used deep residual networks and a transfer learning method to identify crop pests under complex farmland background. Shen et al. [9] developed an improved inception network to extract features and exploited Faster R-CNN to detect insect areas. Pre-trained models can be easily accessed on the internet [10]. It is unnecessary to make massive adjustments to them to get high classification accuracy [11], which facilitates the widespread use of deep CNNs. However, most pre-trained models, such as VGG, ResNet, and Inception, were designed based on ImageNet [12]. Overdependence on them without considering the actual scale of the target image dataset can result in a waste of computational resources.

Feature reuse is an indispensable component of many deep network architectures. It was first employed in the Highway Network [13] which defined two gating units to express how much output is produced by transforming and carrying input. ResNet [14] simplified the reuse method by adding input features directly to output features. DenseNet [15] boosted the frequency of skip-connections. Each layer in it can receive feature maps of all the preceding layers as inputs. This dense connectivity pattern improves the flow of information and gradients throughout the network. However, it also increases the probability of reusing low-level features that contribute little to classification accuracy [16]. The concatenation operation employed by DenseNet was still followed in this paper. Meanwhile, to reduce the effect of dense connections on parameter efficiency, only short-distance connections between layers were retained. It was proven that the classification layer of CNN models focuses on exploiting higher-level features [15,17]. We also observed that the utilization of different hierarchical features by the classification layer varies with the scale of the object in the image. Based on these facts, we propose a new feature reuse method that adapts to scale changes.

A new citrus pest and disease image dataset was recently created [18]. The images in the dataset were collected from a tangerine orchard on Jeju Island and open databases on the internet. Our model and several other benchmark networks were selected to recognize them. The proposed BridgeNet achieved higher classification accuracy than our previous model (Weakly DenseNet [18]), which proves the effectiveness of the new feature reuse method. Compared with training from scratch, ImageNet pre-training sped up convergence and improved performance. However, the network architectures designed for ImageNet demonstrated very serious over-fitting problems without using the transfer learning method [18], which indicates that they are over-complicated in fitting this small-scale image dataset.

The rest of this paper is organized as follows: Section 2 reviews the related work. Section 3 introduces the citrus pest and disease image dataset. Section 4 details the proposed CNN model. Comparison results and analysis are provided in Section 5, and Section 6 presents the conclusion.

## 2. Related Work

Compared with traditional machine learning methods, deep learning is an end-to-end model which does not require the design of handcrafted features for classifiers [19]. In addition, the development of computer hardware allows researchers to train networks with more hidden layers. Due to these advantages, deep learning became a popular data analysis method in many research fields. Esteva et al. [20] used the GoogleNet Inception v3 CNN architecture to classify skin cancer images. The model achieved dermatologist-level accuracy. Dorafshan et al. [21] employed four common edge-detection methods and AlexNet to detect cracks in concrete. Their experimental results showed that the deep CNN model performed much better than other approaches. Alves et al. [22] adopted deep residual networks to identify cotton pests. They provided a field-based image dataset containing 1600 images of 15 pests. The improved ResNet-34 achieved the highest classification accuracy. Chen et al. [23] created a CNN architecture named INC-VGGN to detect rice and maize diseases. They replaced the last few layers of VGG-19 with two Inception modules to further enhance the feature extraction ability. Compared with original networks, this new model showed the best performance. We designed a new feature reuse block (bridge connection) to adapt to the scale changes of citrus pests and diseases in images.

Network architecture has a considerable effect on model performance. In general, a deeper network constructed results in a higher accuracy being delivered [24]. However, as depth increases, training becomes more difficult. To alleviate this problem, different feature reuse methods were proposed [13,14,15]. Inception [25] and Fractal Networks [26] provide sub-paths of different lengths for each building unit. They can be designed to be very deep without needing skip-connections. More recently, the application of attention mechanisms was regarded as another efficient way to improve network precision [27,28,29]. These attention mechanisms can help a model focus on using salient features. These network architectures achieved great success in large-scale image classification tasks.

ImageNet pre-training enables people to easily access encouraging results [11]. For this reason, models designed for ImageNet are frequently used without considering the characteristics of the target dataset. Recht et al. [30] proved that the accuracy of the same network will show a large drop on a new dataset that has a similar distribution to the original data. This phenomenon indicates that any network architecture should be optimized when facing new datasets.

## 3. Image Dataset Description

Our dataset has 12,561 images. It covers 17 species of citrus pest and seven types of citrus disease. Each class contains over 350 images. Figure 1 shows the number of samples in each category. The image collection methods for citrus pests and diseases were described in our previous research [18]. To balance the data distribution in the dataset and improve the model’s generalization ability, several data augmentation approaches were adopted. Table 1 depicts the parameter settings for each data augmentation. Instead of using only one operation, we randomly selected three operations and performed them sequentially to produce a new image. This method can considerably increase the diversity of generated images (Figure 2).

Image samples used in earlier studies were gathered under laboratory conditions [6,31], which reduced the robustness of the trained model to realistic conditions [32]. In contrast, for this study, we collected images with variable, realistic backgrounds. In addition, to further adapt to real-world scenes, the original image was not excessively cropped to only keep the target object region [33]. Figure 3 shows images with varying distances between the camera sensor and the citrus pest or disease.

## 4. Network Architecture

Computer vision competitions greatly promoted the development of deep learning. Many advanced design methods were proposed to improve network performance, which changed the situation of simply stacking convolutional layers. We followed the strategy of SqueezeNet [34] to design the network structure from micro to macro. To save computational cost, our network depth was gradually increased until the accuracy was not significantly improved.

### 4.1. Microstructure of Building Unit

Attention mechanisms usually produce an attention map to highlight the important features, which brings an additional computation overhead and increases the optimization difficulty. We followed the micro-construction of the Network in Network [35] to enhance the features generated by each 3 × 3 convolution (Figure 4). This structure is compatible with the mainframe of a network with no need for extra branches. Furthermore, the Mlpconv layer receives each whole feature map as an input, avoiding over-compressing information like SE (Squeeze-and-Excitation) blocks [27].

### 4.2. Macro Connection between Building Blocks

In order to address the degradation problem, He et al. [14] developed a residual learning framework to add input features to output features. Huang et al. [15] adopted a concatenation operation to increase the frequency of feature reuse. Compared with the add operation, the concatenation feature reuse method is easier to use, which does not require a 1 × 1 convolution to align input and output channels [34]. In addition, concatenation takes less computation time than element-wise addition [17]. For these reasons, we reused previous layer features using concatenation. ShuffleNet V2 [17] proved that the amount of feature reuse decayed exponentially with the distance between two blocks. To avoid introducing redundancy, we only established connections between adjacent layers (Figure 5).

### 4.3. Adaption to Object Scale in the Image

It is well known that CNN models are very sensitive to translations and rotations [36]. We observed that scale changes to an object in an image can also affect neural network performance (Figure 6). In order to find the reason for this, we borrowed SE blocks [27] to monitor the contribution of features from different building blocks to the classification (Figure 7). After training, we selected several groups of images to validate the feature importance distribution in each SE block (Figure 8). For easy comparison, we divided features into three levels based on their importance values. Table 2 presents the number of features in each level. It can be seen that, as the object scale is reduced, the number of high-level features decreases while the number of mid-level features increases. This means that a network has to use more mid-level features to identify the class of smaller-size objects in images.

There are many mid-level and high-quality features in the intermediate layers. However, direct reuse of them will increase the computational complexity of the classification layer. In addition, the proportion of low-level features in shallower layers was greater than in deeper layers (Table 2). Based on these facts, we proposed a new feature reuse method to improve parameter efficiency. The 1 × 1 convolution in Figure 9 has two purposes.

Channel compression: To reduce the number of useless features, the number of output channels from the 1 × 1 convolution is fewer than the number of input channels. This function is similar to the transition layer of DenseNet.Feature retention: Unlike a 3 × 3 convolution, a 1 × 1 convolution performs a simple linear transformation, which can largely preserve input feature information. To further strengthen output feature quality, two 1 × 1 convolutions were stacked after the concatenation operation.

Conventional feature reuse strategies (addition and concatenation) do not consider the discrepancy between features from different layers. More specifically, the difference between shallow features and deep features is not only in the distribution characteristic, but also in the representation complexity. This complexity difference between adjacent layers can be measured by Equation (1). As the distance between layers increases, the difference mentioned will be more significant. The progressive feature reuse method shown in Figure 9 ensures a strong correlation between concatenated features. In addition, the two 1 × 1 convolutions in each feature reuse block reduce the complexity difference between features from long-distance layers.
(1)$$F_{n}^{i}\;=\;\sum_{j=1}^{k}\left(F_{n-1}^{j}*\;W_{j}\;+\;b_{j}\right),$$
where $F_{n}^{i}$ represents the *i*-th feature map of the *n*-th layer, $W_{j}$ and $b_{j}$ denote the corresponding convolutional kernel and bias to the $F_{n-\;1}^{j}$ feature map, and k presents the number of feature maps in the *(n−1)*-th layer.

## 5. Experiments and Results

### 5.1. Experiment Preparation

The overall architecture of BridgeNet for citrus pests and disease recognition is shown in Table 3. Several baseline networks and their variants that have a similar depth to BridgeNet were selected for comparison. We replaced the two 1 × 1 convolutions of the Mlpconv block with CBAM (Convolutional Block Attention Module) [28] to compare their performance. In addition, the bridge connection was compared with deformable convolution. We followed the suggestion of Dai et al. [29] to apply these deformable convolutions in the last three convolutional layers (with kernel size > 1). All these networks shared the same classification block (Figure 10) and were trained with identical optimization schemes. Before training, the original image dataset was divided into a training set, a validation set, and a test set in the ratio of 4:1:1. Then, each model was trained and tested based on them. The three parts did not contain the same samples, and data augmentation was performed for each model and only on the training set. We saved the models that had the highest validation accuracy and examined their generalization ability on the test set. The model hyper-parameters presented in Table 4 were determined by trial and error.

### 5.2. Classification Performance

Table 5 displays the classification accuracy for each model. BridgeNet-19 achieved the highest validation accuracy, followed by VGG-19 with bridge connections and then pre-trained VGG-19. The test accuracy of the models followed the same trend as the validation accuracy, except that the pre-trained VGG-19 ranked second, followed by VGG-19 with bridge connections. Obviously, the models trained from scratch produced lower accuracy than their ImageNet pre-trained counterparts. However, the pre-trained models consumed much more computing resources than their competitors.

It is of note that the use of deformable convolutions does not improve VGG-16 performance. In contrast, the application of bridge connections considerably increased validation and test accuracy. As for additional computational cost, the bridge connection created much less of a computational burden than deformable convolution; the use of bridge connections in the model costed 5.8 MB and the cost of using deformable convolutions was 40.8 MB. Weakly DenseNet-19 performed better than CBAMNet, which indicates that the two 1 × 1 convolutional layers used for feature enhancement are more effective than the attention mechanism. The accuracy of Weakly DenseNet-19 was further increased by using features from the middle layers for classification. In terms of additional computational cost, BridgeNet-19 spent 12.3 MB and MSN-19 consumed 10.3 MB. However, BridgeNet-19 obtained better performance than MSN-19, proving the higher parameter efficiency of using bridge connections. The smaller-size models took less training time per batch except for MSN-19, which had a slower training speed than BridgeNet-19. Figure 11 depicts the training details of each model. Models trained with ImageNet pre-training displayed the fastest convergence. However, models with branch structures required more epochs to reach the final convergence state. This phenomenon indicates that simpler structural models are easier to train.

To validate the effectiveness of the new feature reuse method for adapting to object scale changes, the confusion matrices of Weakly DenseNet-19 and BridgeNet-19 were compared (Figure 12). Using the comparison results, Figure 13 presents the images that were correctly identified by BridgeNet-19 but misclassified by Weakly DenseNet-19. It can be seen that, with the help of bridge connections, BridgeNet-19 has an enhanced ability to correctly classify images with small target objects. The use of bridge connections also improves the discrimination of similar categories, for example, the citrus anthracnose and canker (the difference between them is not obvious without close observation).

### 5.3. Ablation Study

We considered the number of bridge connections as a hyper-parameter and explored its impact on network performance. Bridge connections were introduced from top to bottom as the overall number of connections was increased. Weakly DenseNet-19 was used as the backbone architecture. Table 6 reports the comparison results. It can be observed that, as the number of bridge connections increased, the model performance showed initial cumulative growth. When the number was increased to four, the accuracy then decreased. This indicates that excessive use of shallow information will bring more redundancy to the classification layer.

## 6. Conclusion and Future Work

In this study, a new CNN model was developed to identify common pests and diseases in citrus plantations. Each building block of the network contained two 1 × 1 convolutions that were used to enhance the features generated by 3 × 3 convolutional layers. Concatenation operations were used for feature reuse. To reduce redundancy, only features from adjacent layers were concatenated. We observed that, as the size of the target object in the image decreased, the use of mid-level features by the classification layer increased. Using this insight to adapt the model to scale changes, a new feature reuse method called bridge connection was designed. Experimental results show that the proposed BridgeNet-19 achieved the highest classification accuracy (95.47%). Compared with pre-trained models, our network also presented higher parameter efficiency; its model size (68.9 MB) was half of the pre-trained VGG-16 and VGG-19 networks.

Training of deep CNN models usually requires a large-scale image dataset. However, it is very difficult and expensive to collect so many high-quality, close-up sample images in some specific fields such as medicine and biology. Although ImageNet pre-trained models can allow researchers to achieve satisfactory results on different types of datasets quickly and easily, they are designed too bulky and ill-suited to fit small datasets. We hope to find a better solution to solve this problem in the future.

## Figures and Tables

**Figure 1 sensors-20-04992-f001:**
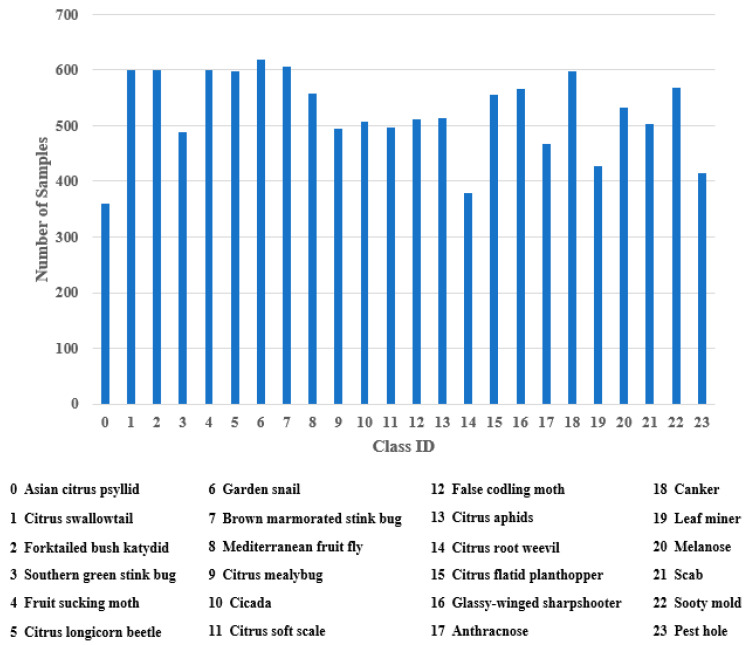
Data distribution of citrus pests and diseases.

**Figure 2 sensors-20-04992-f002:**
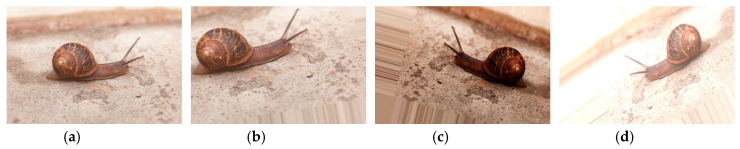
Results of the proposed data augmentation method: (**a**) original image; (**b–d**) generated images.

**Figure 3 sensors-20-04992-f003:**
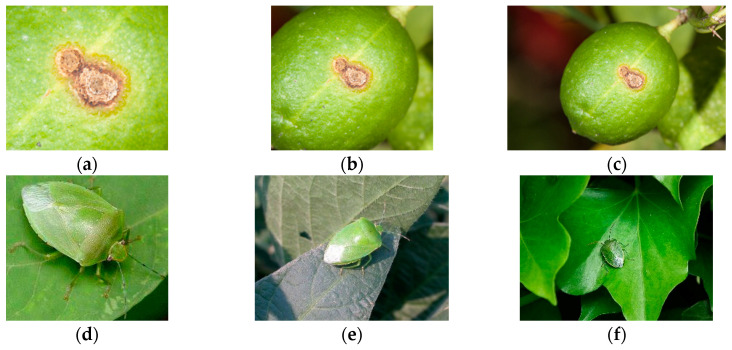
Photos taken with subject at different distances from camera: (**a**–**c**) citrus canker; (**d**–**f**) southern green stink bug.

**Figure 4 sensors-20-04992-f004:**
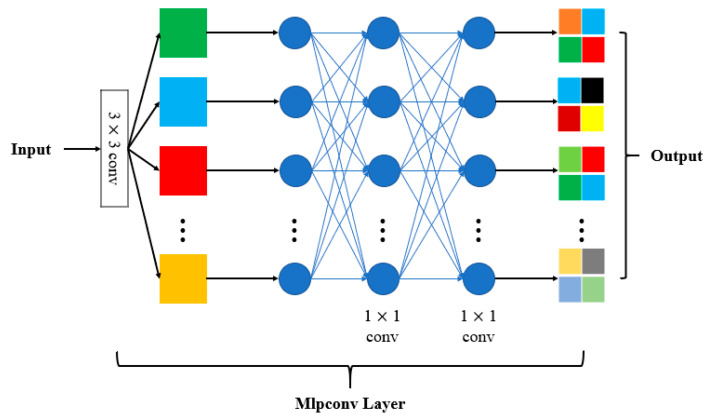
Microstructure of a building block.

**Figure 5 sensors-20-04992-f005:**
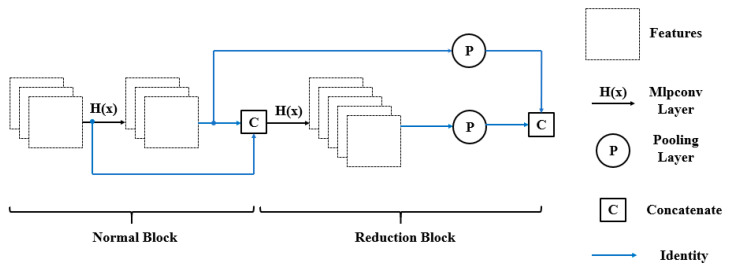
Connection across building blocks.

**Figure 6 sensors-20-04992-f006:**
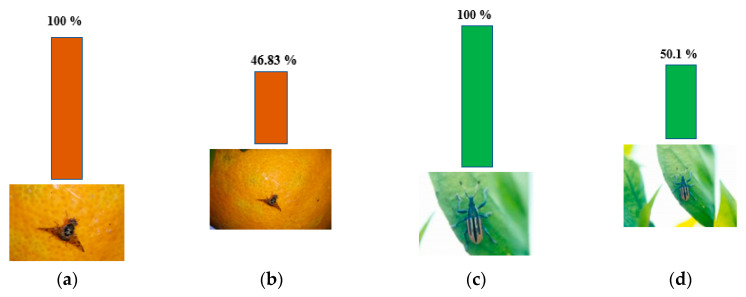
Prediction accuracy of Weakly DenseNet-19 for the different scales of target objects: (**a**,**b**) fruit fly; (**c**,**d**) root weevil.

**Figure 7 sensors-20-04992-f007:**
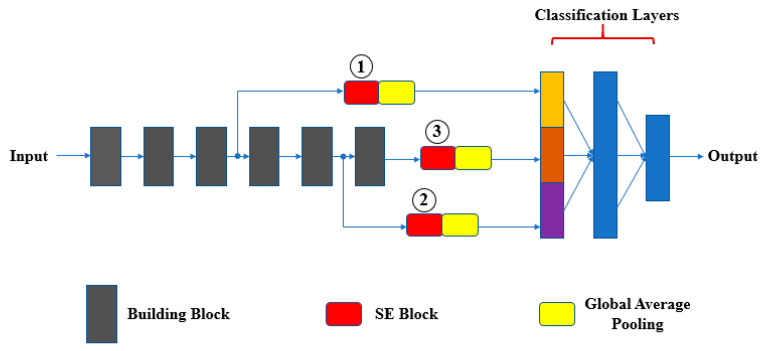
Application of SE (Squeeze-and-Excitation) blocks in testing the importance of middle layer features for classification. This network architecture is called Multi-Scale-Net (MSN) and is compared with other benchmark networks in Section 5.

**Figure 8 sensors-20-04992-f008:**
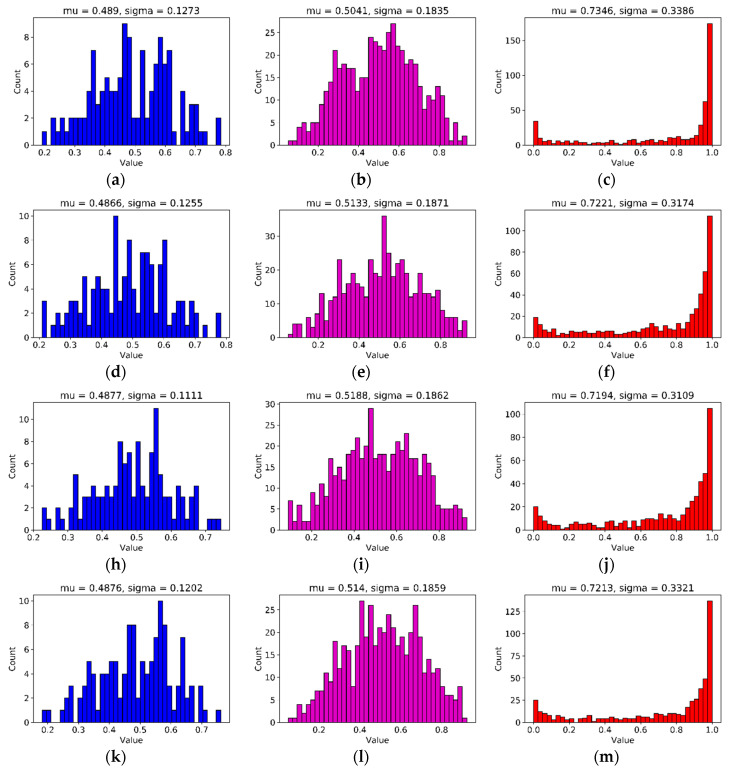
Visualization of each SE block; **mu** and **sigma** represent mean and standard deviation. (**a–c**) and (**k–m**) are feature importance distributions for large-size canker (Figure 3a) and large-size southern green stink bug (Figure 3d); (**d–f**) and (**n–p**) are feature importance distributions for middle-size canker (Figure 3b) and middle-size southern green stink bug (Figure 3e); (**h–j**) and (**q–s**) are feature importance distributions for small-size canker (Figure 3c) and small-size southern green stink bug (Figure 3f); (**a**), (**d**), (**h**), (**k**), (**n**), and (**q**) are the output of SE block 1; (**b**), (**e**), (**i**), (**l**), (**o**), and (**r**) are the output of SE block 2; (**c**), (**f**), (**j**), (**m**), (**p**), and (**s**) are the output of SE block 3.

**Figure 9 sensors-20-04992-f009:**
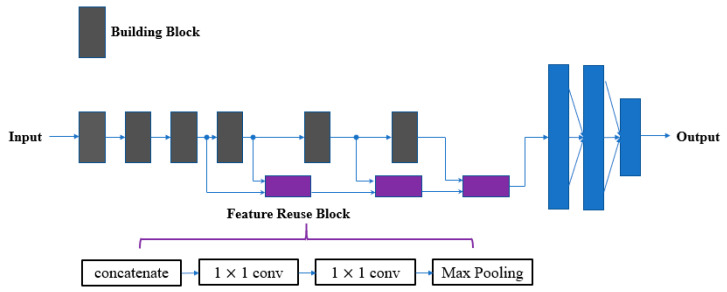
Proposed feature reuse method.

**Figure 10 sensors-20-04992-f010:**
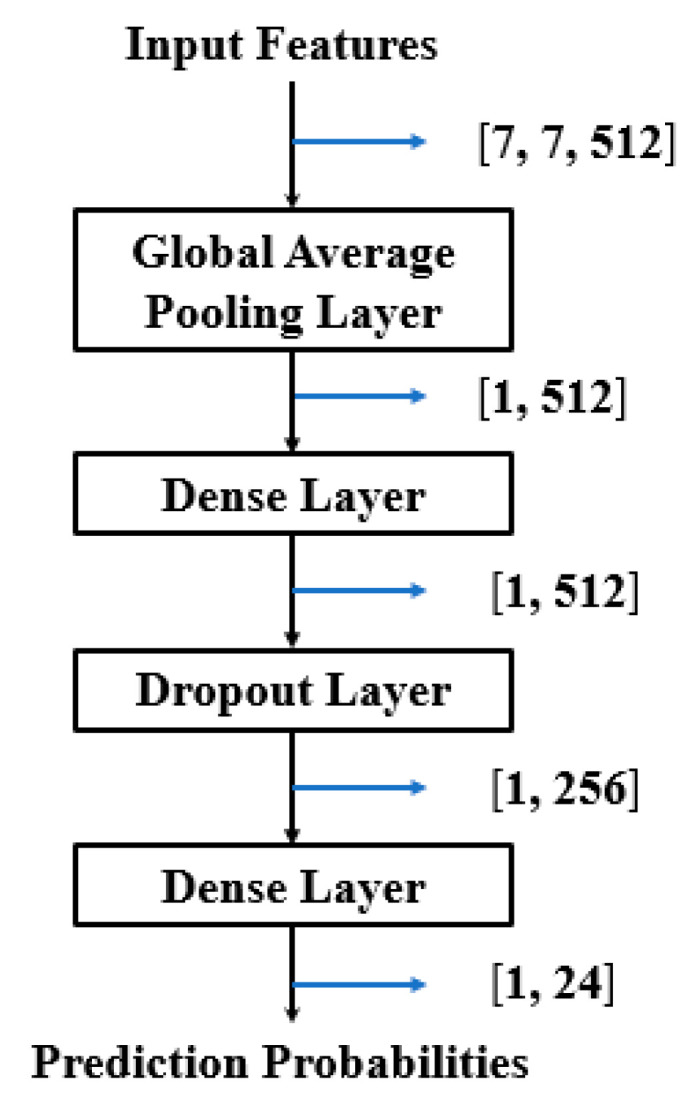
Classification block for each model.

**Figure 11 sensors-20-04992-f011:**
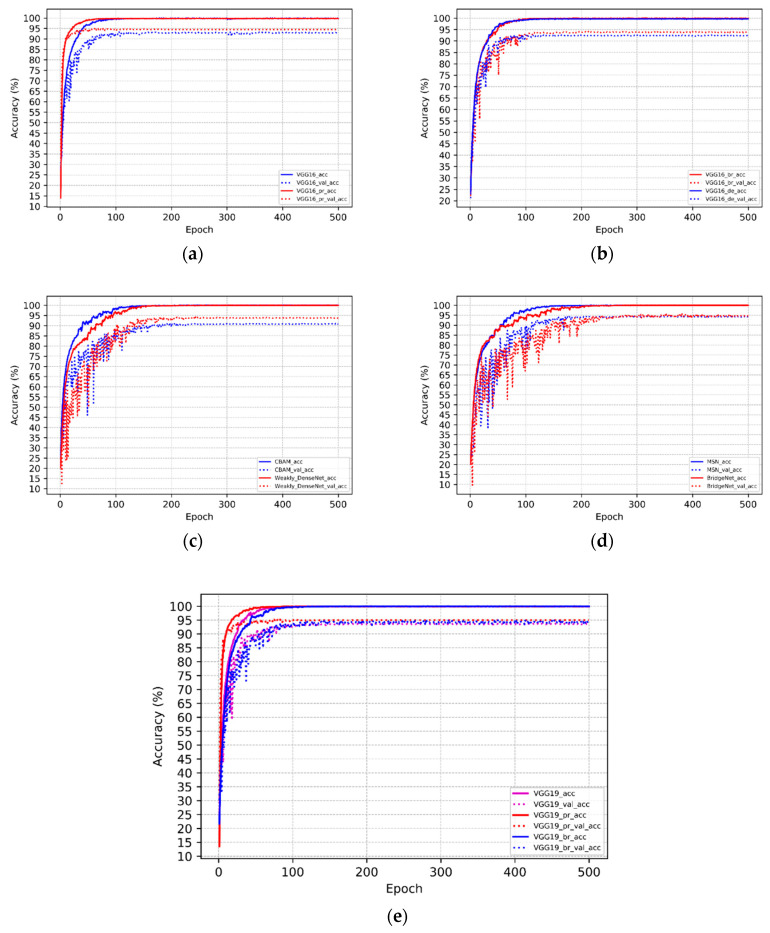
Training process of selected models: “—” denotes training accuracy; “…” represents validation accuracy. (**a**) Red lines are VGG-16 trained with ImageNet pre-training, and blue lines are VGG-16 trained from scratch; (**b**) red lines are VGG-16 with bridge connections, and blue lines are VGG-16 with deformable convolutions; (**c**) red lines are Weakly DenseNet, and blue lines are CBAMNet; (**d**) red lines are BridgeNet-19, and blue lines are MSN-19; (**e**) red lines are pre-trained VGG-19, blue lines are VGG-19 with bridge connections, and magenta lines are VGG-19 trained from scratch.

**Figure 12 sensors-20-04992-f012:**
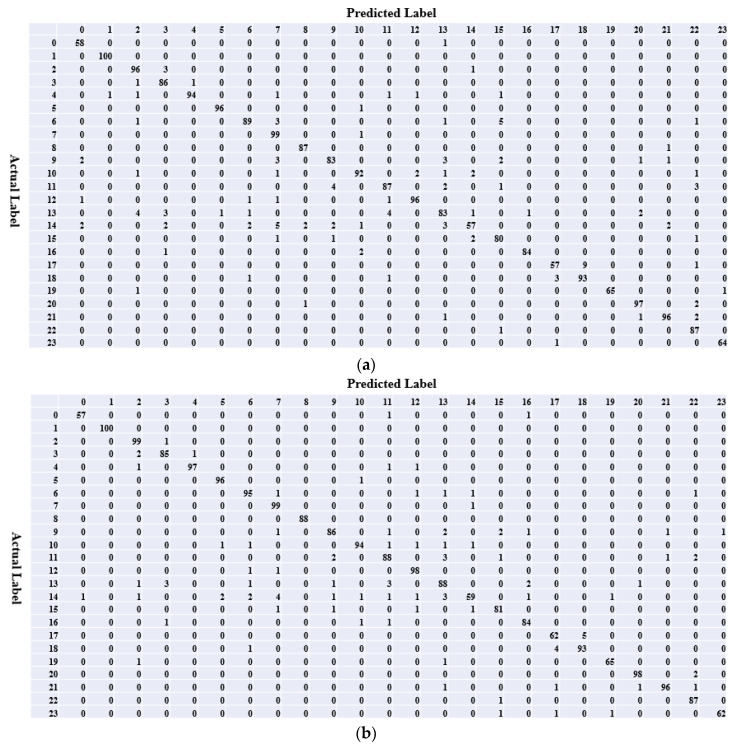
Comparison of confusion matrix: (**a**) Weakly DenseNet-19; (**b**) BridgeNet-19.

**Figure 13 sensors-20-04992-f013:**
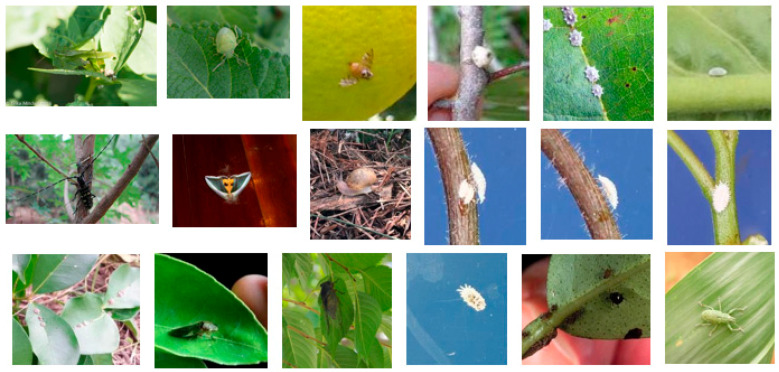
Examples of images misclassified by Weakly DenseNet-19: only images that were misclassified due to the small subject size are presented.

**Table 1 sensors-20-04992-t001:** Data augmentation operations.

Operation Name.	Value
Rotation	[0°, 30°]
Width shift	[0, 0.2]
Height shift	[0, 0.2]
Zoom	[0.8, 1.2]
Horizontal flip	-
Brightness	[0.6, 1.4]

**Table 2 sensors-20-04992-t002:** Statistics for different levels of features.

		**The Number of Low-Level Features** ** (Value ** < ** 0.6)**	**The Number of Mid-Level Features (0.6 ** ≤ ** Value ** < ** 0.95)**	**The Number of High-Level Features ** **(Value ** ≥ ** 0.95)**
Large-size canker	SE block 1	102	26	0
	SE block 2	350	162	0
	SE block 3	138	138	236
	Total	**590**	**326**	**236**
Middle-size canker	SE block 1	104	24	0
	SE block 2	343	169	0
	SE block 3	139	198	175
	Total	**586**	**391**	**175**
Small-size canker	SE block 1	108	20	0
	SE block 2	326	186	0
	SE block 3	137	222	153
	Total	**571**	**428**	**153**
Large-size green stink bug	SE block 1	107	21	0
	SE block 2	334	178	0
	SE block 3	142	184	186
	Total	**583**	**383**	**186**
Middle-size green stink bug	SE block 1	107	21	0
	SE block 2	336	176	0
	SE block 3	147	208	157
	Total	**590**	**405**	**157**
Small-size green stink bug	SE block 1	103	25	0
	SE block 2	323	189	0
	SE block 3	143	238	131
	Total	**569**	**452**	**131**

**Table 3 sensors-20-04992-t003:** Structure of BridgeNet-19. The input size is 224 × 224 × 3. The initial block followed the setting of ResNet-50.

Building Unit		Output Dimension	
Initial Block		56 × 56 × 32	
Normal Block		56 × 56 × 96	
Reduction Block		28 × 28 × 192	
Normal Block		28 × 28 × 384	
Reduction Block	Feature Reuse Block	14 × 14 × 768	14 × 14 × 256
Mlpconv Layer with Max Pooling	Feature Reuse Block	7 × 7 × 512	7 × 7 × 512
Feature Reuse Block without Max Pooling		7 × 7 × 512	
Classification Block		1 × 24	

**Table 4 sensors-20-04992-t004:** Hyper-parameters for training models.

Model Hyper-Parameter	Method/Value
Weight initialization	He normal [37]
Batch size	16
Weight decay	L2/0.005 [38]
Optimizer	SGD
Momentum	Nesterov/0.9 [39]
Dropout rate	0.5 [40]
Initial learning rate	Training from scratch/0.001
	ImageNet pre-training/0.0001
Learning rate schedule	Adaptive [18]
Training epoch	500

**Table 5 sensors-20-04992-t005:** Comparison of model performance.

Model Name	Training Accuracy (%)	Validation Accuracy (%)	Test Accuracy (%)	Model Size (MB)	Training Speed ms/Batch Size
VGG-16	99.82	93	92.93	120.2	303
Pre-trained VGG-16	99.79	94.96	94.45	120.2	303
VGG-16 with bridge connections	99.89	94.17	93.89	126	343
VGG-16 with deformable convolutions	99.59	92.63	92.38	161	635
VGG-19	99.89	93.8	93.57	155	352
Pre-trained VGG-19	99.91	95.2	95.01	155	352
VGG-19 with bridge connections	99.93	95.33	94.73	167	392
Weakly DenseNet-19	99.79	94.03	93.71	56.6	139
CBAMNet	**99.97**	91.42	90.79	**49.5**	**139**
MSN-19	99.8	94.5	94.24	66.9	142
BridgeNet-19	**99.97**	**95.61**	**95.47**	68.9	140

Implementation of models is available at https://github.com/xingshulicc/xingshulicc/tree/master/citrus_pest_diseases_recognition.

**Table 6 sensors-20-04992-t006:** Effect of using a different number of bridge connections.

Number	Test Error (%)	Model Size (MB)
0	6.29	56.6
2	4.57	67.6
3	4.53	68.9
4	4.62	69.3

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
