# Peer review of "Classification Accuracy Improvement for Small-Size Citrus Pests and Diseases Using Bridge Connections in Deep Neural Networks"

_sensors, 2020, doi:10.3390/s20174992_

Round 1
Reviewer 1 Report
Summary:
The classification performance of convolutional neural networks degrade as the size of the target object in the image decreased. To adapt to scale changes, this paper propose a new feature reuse method named bridge connection to concat with both the large, middle and small size features. With the help of bridge connections, the accuracy of baseline networks was improved at little additional computation cost.
Strength:
1. The author borrow SE blocks to monitor the contribution of features from different building blocks to the classification, to validate the feature importance distribution under different size objects. The result show that as the object scale is reduced, the number of high-level features decreases while the number of mid-level features increases.
2. It proposed a new feature reuse block to improve parameter efficiency with two 1 脳 1 convolutions, which reduce the number of useless features and preserve features in each level.
Weakness:
1. The results presented in Table 6 seems just the highest accuracy of model. Due to the learning process of the network is not controllable, the mean accuracy of several training models maybe more convinced.
2. The paper lack some experiment details. Does image dataset is divided for each model randomly and data augmentation is performed for each method? The paper demonstrate the method on the one dataset, does any other image dataset can also be used to prove the effective?
Comments for the authors:
This paper have a detail discussion about network architecture in classification task. The use of SE blocks to monitor the contribution of features from different building blocks to the classification and the analysis for the number of features in each level are convincing. And the improved bridge connection method based on the analysis results is proved to be effective by experiments. This paper is also clearly structured. I expect more experiment details and other dataset can be show in the further version.
Reviewer 2 Report
1) The statement in lines 53-54 is crucial, please elaborate more and use references.
2) The statement in lines 75-76 is crucial too, please provide evidence to this claim.
3) The dataset is not balanced (Asian citrus psyllid
359, while Garden snail 618). This causes the classifier to bias towards the class that has significantly more samples than other classes. Please verify.
4) Please elaborate more on the additional computation cost and compute the O(c) "Big O".
5) Please include the inference time (latency) in milliseconds (ms) for the proposed model compared to other models.
Reviewer 3 Report
Related work section should be improved. This section do not has relevant references to pest detection and classification topic.
in line 135, I do not understand the sentence " To save computational cost, our network depth was gradually increased until the accuracy was not significantly improved.", can you please explain? (maybe you mean until accuracy be acceptable? )
section 3. Image Dataset Description, can be shorted, no need of table 1.
It will be great contribution for this article, if there is public link with bridgenet-19 model
Round 2
Reviewer 2 Report
Please incorporate the responses in the manuscript.
Author Response
Dear reviewer,
Thank you for your suggestion. We have annotated the changes in the manuscript. The revised version about this paper has been uploaded again. More explanations about the data augmentation have been shown in the first round revision in the attachment. And additional computation cost information has been added. Thank you again.